# Following the Leader and Fast Rates in Linear Prediction: Curved Constraint Sets and Other Regularities

**Ruitong Huang**
Department of Computing Science
University of Alberta, AB, Canada
ruitong@ualberta.ca

**Tor Lattimore**
School of Informatics and Computing
Indiana University, IN, USA
tor.lattimore@gmail.com

**András György**
Dept. of Electrical & Electronic Engineering
Imperial College London, UK
a.gyorgy@imperial.ac.uk

**Csaba Szepesvári**
Department of Computing Science
University of Alberta, AB, Canada
szepesva@ualberta.ca

## Abstract

The follow the leader (FTL) algorithm, perhaps the simplest of all online learning algorithms, is known to perform well when the loss functions it is used on are positively curved. In this paper we ask whether there are other "lucky" settings when FTL achieves sublinear, "small" regret. In particular, we study the fundamental problem of linear prediction over a non-empty convex, compact domain. Amongst other results, we prove that the curvature of the boundary of the domain can act as if the losses were curved: In this case, we prove that as long as the mean of the loss vectors have positive lengths bounded away from zero, FTL enjoys a logarithmic growth rate of regret, while, e.g., for polyhedral domains and stochastic data it enjoys finite expected regret. Building on a previously known meta-algorithm, we also get an algorithm that simultaneously enjoys the worst-case guarantees and the bound available for FTL.

## 1 Introduction

Learning theory traditionally has been studied in a statistical framework, discussed at length, for example, by Shalev-Shwartz and Ben-David [2014]. The issue with this approach is that the analysis of the performance of learning methods seems to critically depend on whether the data generating mechanism satisfies some probabilistic assumptions. Realizing that these assumptions are not necessarily critical, much work has been devoted recently to studying learning algorithms in the so-called online learning framework [Cesa-Bianchi and Lugosi, 2006]. The online learning framework makes minimal assumptions about the data generating mechanism, while allowing one to replicate results of the statistical framework through online-to-batch conversions [Cesa-Bianchi et al., 2004]. By following a minimax approach, however, results proven in the online learning setting, at least initially, led to rather conservative results and algorithm designs, failing to capture how more regular, "easier" data, may give rise to faster learning speed. This is problematic as it may suggest overly conservative learning strategies, missing opportunities to extract more information when the data is nicer. Also, it is hard to argue that data resulting from passive data collection, such as weather data, would ever be adversarially generated (though it is equally hard to defend that such data satisfies precise stochastic assumptions). Realizing this issue, during recent years much work has been devoted to understanding what regularities and how can lead to faster learning speed. For example, much work has been devoted to showing that faster learning speed (smaller "regret") can be achieved in

the online convex optimization setting when the loss functions are "curved", such as when the loss functions are strongly convex or exp-concave, or when the losses show small variations, or the best prediction in hindsight has a small total loss, and that these properties can be exploited in an adaptive manner (e.g., Merhav and Feder 1992, Freund and Schapire 1997, Gaivoronski and Stella 2000, Cesa-Bianchi and Lugosi 2006, Hazan et al. 2007, Bartlett et al. 2007, Kakade and Shalev-Shwartz 2009, Orabona et al. 2012, Rakhlin and Sridharan 2013, van Erven et al. 2015, Foster et al. 2015).

In this paper we contribute to this growing literature by studying online linear prediction and the follow the leader (FTL) algorithm. Online linear prediction is arguably the simplest of all the learning settings, and lies at the heart of online convex optimization, while it also serves as an abstraction of core learning problems such as prediction with expert advice. FTL, the online analogue of empirical risk minimization of statistical learning, is the simplest learning strategy, one can think of. Although the linear setting of course removes the possibility of exploiting the curvature of losses, as we will see, there are multiple ways online learning problems can present data that allows for small regret, even for FTL. As is it well known, in the worst case, FTL suffers a linear regret (e.g., Example 2.2 of Shalev-Shwartz [2012]). However, for "curved" losses (e.g., exp-concave losses), FTL was shown to achieve small (logarithmic) regret (see, e.g., Merhav and Feder [1992], Cesa-Bianchi and Lugosi [2006], Gaivoronski and Stella [2000], Hazan et al. [2007]).

In this paper we take a thorough look at FTL in the case when the losses are linear, but the problem perhaps exhibits other regularities. The motivation comes from the simple observation that, for prediction over the simplex, when the loss vectors are selected independently of each other from a distribution with a bounded support with a nonzero mean, FTL quickly locks onto selecting the loss-minimizing vertex of the simplex, achieving finite expected regret. In this case, FTL is arguably an excellent algorithm. In fact, FTL is shown to be the minimax optimizer for the binary losses in the stochastic expert setting in the paper of Kotłowski [2016]. Thus, we ask the question of whether there are other regularities that allow FTL to achieve nontrivial performance guarantees. Our main result shows that when the decision set (or constraint set) has a sufficiently "curved" boundary and the linear loss is bounded away from 0, FTL is able to achieve logarithmic regret even in the adversarial setting, thus opening up a new way to prove fast rates based on not on the curvature of losses, but on that of the boundary of the constraint set and non-singularity of the linear loss. In a matching lower bound we show that this regret bound is essentially unimprovable. We also show an alternate bound for polyhedral constraint sets, which allows us to prove that (under certain technical conditions) for stochastic problems the expected regret of FTL will be finite. To finish, we use $(\mathcal{A}, \mathcal{B})$-prod of Sani et al. [2014] to design an algorithm that adaptively interpolates between the worst case $O(\sqrt{n})$ regret and the smaller regret bounds, which we prove here for "easy data." Simulation results on artificial data to illustrate the theory complement the theoretical findings, though due to lack of space these are presented only in the long version of the paper [Huang et al., 2016].

While we believe that we are the first to point out that the curvature of the constraint set $\mathcal{W}$ can help in speeding up learning, this effect is known in convex optimization since at least the work of Levitin and Polyak [1966], who showed that exponential rates are attainable for strongly convex constraint sets if the norm of the gradients of the objective function admit a uniform lower bound. More recently, Garber and Hazan [2015] proved an $O(1/n^2)$ optimization error bound (with problem-dependent constants) for the Frank-Wolfe algorithm for strongly convex and smooth objectives and strongly convex constraint sets. The effect of the shape of the constraint set was also discussed by Abbasi-Yadkori [2010] who demonstrated $O(\sqrt{n})$ regret in the linear bandit setting. While these results at a high level are similar to ours, our proof technique is rather different than that used there.

## 2 Preliminaries, online learning and the follow the leader algorithm

We consider the standard framework of online convex optimization, where a learner and an environment interact in a sequential manner in $n$ rounds: In round every round $t = 1, \ldots, n$, first the learner predicts $w_t \in \mathcal{W}$. Then the environment picks a loss function $\ell_t \in \mathcal{L}$, and the learner suffers loss $\ell_t(w_t)$ and observes $\ell_t$. Here, $\mathcal{W}$ is a non-empty, compact convex subset of $\mathbb{R}^d$ and $\mathcal{L}$ is a set of convex functions, mapping $\mathcal{W}$ to the reals. The elements of $\mathcal{L}$ are called loss functions. The performance of the learner is measured in terms of its regret,

$$R_n = \sum_{t=1}^{n} \ell_t(w_t) - \min_{w \in \mathcal{W}} \sum_{t=1}^{n} \ell_t(w).$$

The simplest possible case, which will be the focus of this paper, is when the losses are linear, i.e., when $\ell_t(w) = \langle f_t, w \rangle$ for some $f_t \in \mathcal{F} \subset \mathbb{R}^d$. In fact, the linear case is not only simple, but is also fundamental since the case of nonlinear loss functions can be reduced to it: Indeed, even if the losses are nonlinear, defining $f_t \in \partial \ell_t(w_t)$ to be a subgradient[1] of $\ell_t$ at $w_t$ and letting $\tilde{\ell}_t(u) = \langle f_t, u \rangle$, by the definition of subgradients, $\ell_t(w_t) - \ell_t(u) \leq \ell_t(w_t) - (\ell_t(w_t) + \langle f_t, u - w_t \rangle) = \tilde{\ell}_t(w_t) - \tilde{\ell}_t(u)$, hence for any $u \in \mathcal{W}$,

$$\sum_t \ell_t(w_t) - \sum_t \ell_t(u) \leq \sum_t \tilde{\ell}_t(w_t) - \sum_t \tilde{\ell}_t(u) \,.$$

In particular, if an algorithm keeps the regret small no matter how the linear losses are selected (even when allowing the environment to pick losses based on the choices of the learner), the algorithm can also be used to keep the regret small in the nonlinear case. Hence, in what follows we will study the linear case $\ell_t(w) = \langle f_t, w \rangle$ and, in particular, we will study the regret of the so-called "Follow The Leader" (FTL) learner, which, in round $t \geq 2$ picks

$$w_t = \operatorname*{argmin}_{w \in \mathcal{W}} \sum_{i=1}^{t-1} \ell_i(w) \,.$$

For the first round, $w_1 \in \mathcal{W}$ is picked in an arbitrary manner. When $\mathcal{W}$ is compact, the optimal $w$ of $\min_{w \in \mathcal{W}} \sum_{i=1}^{t-1} \langle w, f_t \rangle$ is attainable, which we will assume henceforth. If multiple minimizers exist, we simply fix one of them as $w_t$. We will also assume that $\mathcal{F}$ is non-empty, compact and convex.

## 2.1 Support functions

Let $\Theta_t = -\frac{1}{t} \sum_{i=1}^t f_i$ be the negative average of the first $t$ vectors in $(f_t)_{t=1}^n$, $f_t \in \mathcal{F}$. For convenience, we define $\Theta_0 := 0$. Thus, for $t \geq 2$,

$$w_t = \operatorname*{argmin}_{w \in \mathcal{W}} \sum_{i=1}^{t-1} \langle w, f_i \rangle = \operatorname*{argmin}_{w \in \mathcal{W}} \langle w, -\Theta_{t-1} \rangle = \operatorname*{argmax}_{w \in \mathcal{W}} \langle w, \Theta_{t-1} \rangle \,.$$

Denote by $\Phi(\Theta) = \max_{w \in \mathcal{W}} \langle w, \Theta \rangle$ the so-called *support function* of $\mathcal{W}$. The support function, being the maximum of linear and hence convex functions, is itself convex. Further $\Phi$ is positive homogenous: for $a \geq 0$ and $\theta \in \mathbb{R}^d$, $\Phi(a\theta) = a\Phi(\theta)$. It follows then that the epigraph $\mathrm{epi}(\Phi) = \left\{ (\theta, z) \,|\, z \geq \Phi(\theta), z \in \mathbb{R}, \theta \in \mathbb{R}^d \right\}$ of $\Phi$ is a cone, since for any $(\theta, z) \in \mathrm{epi}(\Phi)$ and $a \geq 0$, $az \geq a\Phi(\theta) = \Phi(a\theta)$, $(a\theta, az) \in \mathrm{epi}(\Phi)$ also holds.

The differentiability of the support function is closely tied to whether in the FTL algorithm the choice of $w_t$ is uniquely determined:

**Proposition 2.1.** *Let $\mathcal{W} \neq \emptyset$ be convex and closed. Fix $\Theta$ and let $\mathcal{Z} := \{ w \in \mathcal{W} \,|\, \langle w, \Theta \rangle = \Phi(\Theta) \}$. Then, $\partial \Phi(\Theta) = \mathcal{Z}$ and, in particular, $\Phi(\Theta)$ is differentiable at $\Theta$ if and only if $\max_{w \in \mathcal{W}} \langle w, \Theta \rangle$ has a unique optimizer. In this case, $\nabla \Phi(\Theta) = \operatorname{argmax}_{w \in \mathcal{W}} \langle w, \Theta \rangle$.*

The proposition follows from Danskin's theorem when $\mathcal{W}$ is compact (e.g., Proposition B.25 of Bertsekas 1999), but a simple direct argument can also be used to show that it also remains true even when $\mathcal{W}$ is unbounded.[2] By Proposition 2.1, when $\Phi$ is differentiable at $\Theta_{t-1}$, $w_t = \nabla \Phi(\Theta_{t-1})$.

## 3 Non-stochastic analysis of FTL

We start by rewriting the regret of FTL in an equivalent form, which shows that we can expect FTL to enjoy a small regret when successive weight vectors move little. A noteworthy feature of the next proposition is that rather than bounding the regret from above, it gives an equivalent expression for it.

**Proposition 3.1.** *The regret $R_n$ of FTL satisfies*

$$R_n = \sum_{t=1}^n t \langle w_{t+1} - w_t, \Theta_t \rangle \,.$$

The result is a direct corollary of Lemma 9 of McMahan [2010], which holds for any sequence of losses, even in the lack of convexity. It is also a tightening of the well-known inequality $R_n \leq \sum_{t=1}^{n} \ell_t(w_t) - \ell_t(w_{t+1})$, which again holds for arbitrary loss sequences (e.g., Lemma 2.1 of Shalev-Shwartz [2012]). To keep the paper self-contained, we give an elegant, short direct proof, based on the summation by parts formula:

*Proof.* The summation by parts formula states that for any $u_1, v_1, \ldots, u_{n+1}, v_{n+1}$ reals, $\sum_{t=1}^{n} u_t (v_{t+1} - v_t) = (u_{t+1}v_{t+1} - u_1 v_1) - \sum_{t=1}^{n} (u_{t+1} - u_t) v_{t+1}$. Applying this to the definition of regret with $u_t := w_{t,\cdot}$ and $v_{t+1} := t\Theta_t$, we get $R_n = -\sum_{t=1}^{n} \langle w_t, t\Theta_t - (t-1)\Theta_{t-1} \rangle + \langle w_{n+1}, n\Theta_n \rangle = -\{ \overline{\langle w_{n+1}, n\Theta_n \rangle} - 0 - \sum_{t=1}^{n} \langle w_{t+1} - w_t, t\Theta_t \rangle \} + \overline{\langle w_{n+1}, n\Theta_n \rangle}$. □

Our next proposition gives another formula that is equal to the regret. As opposed to the previous result, this formula is appealing as it is independent of $w_t$; but it directly connects the sequence $(\Theta_t)_t$ to the geometric properties of $\mathcal{W}$ through the support function $\Phi$. For this proposition we will momentarily assume that $\Phi$ is differentiable at $(\Theta_t)_{t \geq 1}$; a more general statement will follow later.

**Proposition 3.2.** *If $\Phi$ is differentiable at $\Theta_1, \ldots, \Theta_n$,*

$$R_n = \sum_{t=1}^{n} t \, D_\Phi(\Theta_t, \Theta_{t-1}) , \tag{1}$$

*where $D_\Phi(\theta', \theta) = \Phi(\theta') - \Phi(\theta) - \langle \nabla\Phi(\theta), \theta' - \theta \rangle$ is the Bregman divergence of $\Phi$ and we use the convention that $\nabla\Phi(0) = w_1$.*

*Proof.* Let $v = \operatorname{argmax}_{w \in \mathcal{W}} \langle w, \theta \rangle$, $v' = \operatorname{argmax}_{w \in \mathcal{W}} \langle w, \theta' \rangle$. When $\Phi$ is differentiable at $\theta$,

$$D_\Phi(\theta', \theta) = \Phi(\theta') - \Phi(\theta) - \langle \nabla\Phi(\theta), \theta' - \theta \rangle = \langle v', \theta' \rangle - \langle v, \theta \rangle - \langle v, \theta' - \theta \rangle = \langle v' - v, \theta' \rangle . \tag{2}$$

Therefore, by Proposition 3.1, $R_n = \sum_{t=1}^{n} t \langle w_{t+1} - w_t, \Theta_t \rangle = \sum_{t=1}^{n} t \, D_\Phi(\Theta_t, \Theta_{t-1})$. □

When $\Phi$ is non-differentiable at some of the points $\Theta_1, \ldots, \Theta_n$, the equality in the above proposition can be replaced with inequalities. Defining the upper Bregman divergence $\overline{D}_\Phi(\theta', \theta) = \sup_{w \in \partial\Phi(\theta)} \Phi(\theta') - \Phi(\theta) - \langle w, \theta' - \theta \rangle$ and the lower Bregman divergence $\underline{D}_\Phi(\theta', \theta)$ similarly with inf instead of sup, similarly to Proposition 3.2, we obtain

$$\sum_{t=1}^{n} t \, \underline{D}_\Phi(\Theta_t, \Theta_{t-1}) \leq R_n \leq \sum_{t=1}^{n} t \, \overline{D}_\Phi(\Theta_t, \Theta_{t-1}) . \tag{3}$$

### 3.1 Constraint sets with positive curvature

The previous results shows in an implicit fashion that the curvature of $\mathcal{W}$ controls the regret. We now present our first main result that makes this connection explicit. Denote the boundary of $\mathcal{W}$ by $\operatorname{bd}(\mathcal{W})$. For this result, we shall assume that $\mathcal{W}$ is $C^2$, that is, $\operatorname{bd}(\mathcal{W})$ is a twice continuously differentiable submanifold of $\mathbb{R}^d$. Recall that in this case the *principal curvatures* of $\mathcal{W}$ at $w \in \operatorname{bd}(\mathcal{W})$ are the eigenvalues of $\nabla u_\mathcal{W}(w)$, where $u_\mathcal{W} : \operatorname{bd}(\mathcal{W}) \to \mathbb{S}^{d-1}$, the so-called Gauss map, maps a boundary point $w \in \operatorname{bd}(\mathcal{W})$ to the unique outer normal vector to $\mathcal{W}$ at $w$.[3] As it is well known, $\nabla u_\mathcal{W}(w)$ is a self-adjoint operator, with nonnegative eigenvalues, thus the principal curvatures are nonnegative. Perhaps a more intuitive, yet equivalent definition, is that the principal eigenvalues are the eigenvalues of the Hessian of $f = f_w$ in the parameterization $t \mapsto w + t - f_w(t) u_\mathcal{W}(w)$ of $\operatorname{bd}(\mathcal{W})$ which is valid in a small open neighborhood of $w$, where $f_w : T_w \mathcal{W} \to [0, \infty)$ is a suitable convex, nonnegative valued function that also satisfies $f_w(0) = 0$ and where $T_w \mathcal{W}$, a hyperplane of $\mathbb{R}^d$, denotes the tangent space of $\mathcal{W}$ at $w$, obtained by taking the support plane $H$ of $\mathcal{W}$ at $w$ and shifting it by $-w$. Thus, the principal curvatures at some point $w \in \operatorname{bd}(\mathcal{W})$ describe the local shape of $\operatorname{bd}(\mathcal{W})$ up to the second order.

A related concept that has been used in convex optimization to show fast rates is that of a strongly convex constraint set [Levitin and Polyak, 1966, Garber and Hazan, 2015]: $\mathcal{W}$ is $\lambda$-strongly convex

with respect to the norm $\|\cdot\|$ if, for any $x, y \in \mathcal{W}$ and $\gamma \in [0, 1]$, the $\|\cdot\|$-ball with origin $\gamma x + (1-\gamma)y$ and radius $\gamma(1-\gamma)\lambda \|x - y\|^2 /2$ is included in $\mathcal{W}$. One can show that a closed convex set $\mathcal{W}$ is $\lambda$-strongly convex with respect to $\|\cdot\|_2$ if and only if the principal curvatures of the surface $\mathrm{bd}\mathcal{W}$ are all at least $\lambda$.

Our next result connects the principal curvatures of $\mathrm{bd}(\mathcal{W})$ to the regret of FTL and shows that FTL enjoys logarithmic regret for highly curved surfaces, as long as $\|\Theta_t\|_2$ is bounded away from zero.

**Theorem 3.3.** *Let $\mathcal{W} \subset \mathbb{R}^d$ be a $C^2$ convex body with $d \geq 2$.[4] Let $M = \max_{f \in \mathcal{F}} \|f\|_2$ and assume that $\Phi$ is differentiable at $(\Theta_t)_t$. Assume that the principal curvatures of the surface $\mathrm{bd}(\mathcal{W})$ are all at least $\lambda_0$ for some constant $\lambda_0 > 0$ and $L_n := \min_{1 \leq t \leq n} \|\Theta_t\|_2 > 0$. Choose $w_1 \in \mathrm{bd}(\mathcal{W})$. Then*

$$R_n \leq \frac{2M^2}{\lambda_0 L_n}(1 + \log(n)).$$

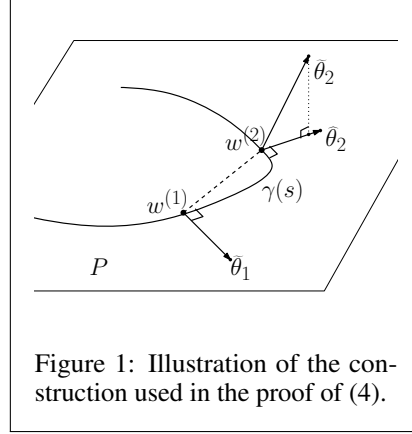

As we will show later in an essentially matching lower bound, this bound is tight, showing that the forte of FTL is when $L_n$ is bounded away from zero and $\lambda_0$ is large. Note that the bound is vacuous as soon as $L_n = O(\log(n)/n)$ and is worse than the minimax bound of $O(\sqrt{n})$ when $L_n = o(\log(n)/\sqrt{n})$. One possibility to reduce the bound's sensitivity to $L_n$ is to use the trivial bound $\langle w_{t+1} - w_t, \Theta_t \rangle \leq LW = L \sup_{w,w' \in \mathcal{W}} \|w - w'\|_2$ for indices $t$ when $\|\Theta_t\| \leq L$. Then, by optimizing the bound over $L$, one gets a data-dependent bound of the form $\inf_{L>0} \left( \frac{2M^2}{\lambda_0 L}(1 + \log(n)) + LW \sum_{t=1}^n t\,\mathbb{I}(\|\Theta_t\| \leq L) \right)$, which is more complex, but is free of $L_n$ and thus reflects the nature of FTL better. Note that in the case of stochastic problems, where $f_1, \dots, f_n$ are independent and identically distributed (i.i.d.) with $\mu := -\mathbb{E}[\Theta_t] \neq 0$, the probability

Figure 1: Illustration of the construction used in the proof of (4).

that $\|\Theta_t\|_2 < \|\mu\|_2 /2$ is exponentially small in $t$. Thus, selecting $L = \|\mu\|_2 /2$ in the previous bound, the contribution of the expectation of the second term is $O(\|\mu\|_2 W)$, giving an overall bound of the form $O(\frac{M^2}{\lambda_0 \|\mu\|_2} \log(n) + \|\mu\|_2 W)$. After the proof we will provide some simple examples that should make it more intuitive how the curvature of $\mathcal{W}$ helps keeping the regret of FTL small.

*Proof.* Fix $\theta_1, \theta_2 \in \mathbb{R}^d$ and let $w^{(1)} = \mathrm{argmax}_{w \in \mathcal{W}} \langle w, \theta_1 \rangle$, $w^{(2)} = \mathrm{argmax}_{w \in \mathcal{W}} \langle w, \theta_2 \rangle$. Note that if $\theta_1, \theta_2 \neq 0$ then $w^{(1)}, w^{(2)} \in \mathrm{bd}(\mathcal{W})$. Below we will show that

$$\langle w^{(1)} - w^{(2)}, \theta_1 \rangle \leq \frac{1}{2\lambda_0} \frac{\|\theta_2 - \theta_1\|_2^2}{\|\theta_2\|_2}. \tag{4}$$

Proposition 3.1 suggests that it suffices to bound $\langle w_{t+1} - w_t, \Theta_t \rangle$. By (4), we see that it suffices to bound how much $\Theta_t$ moves. A straightforward calculation shows that $\Theta_t$ cannot move much:

**Lemma 3.4.** *For any norm $\|\cdot\|$ on $\mathcal{F}$, we have $\|\Theta_t - \Theta_{t-1}\| \leq \frac{2}{t}M$, where $M = \max_{f \in \mathcal{F}} \|f\|$ is a constant that depends on $\mathcal{F}$ and the norm $\|\cdot\|$.*

Combining inequality (4) with Proposition 3.1 and Lemma 3.4, we get

$$R_n = \sum_{t=1}^n t\langle w_{t+1} - w_t, \Theta_t \rangle \leq \sum_{t=1}^n \frac{t}{2\lambda_0} \frac{\|\Theta_t - \Theta_{t-1}\|_2^2}{\|\Theta_{t-1}\|_2}$$

$$\leq \frac{2M^2}{\lambda_0} \sum_{t=1}^n \frac{1}{t\|\Theta_{t-1}\|_2} \leq \frac{2M^2}{\lambda_0 L_n} \sum_{t=1}^n \frac{1}{t} \leq \frac{2M^2}{\lambda_0 L_n}(1 + \log(n)).$$

To finish the proof, it thus remains to show (4).

The following elementary lemma relates the cosine of the angle between two vectors $\theta_1$ and $\theta_2$ to the squared normalized distance between the two vectors, thereby reducing our problem to bounding the cosine of this angle. For brevity, we denote by $\cos(\theta_1, \theta_2)$ the cosine of the angle between $\theta_1$ and $\theta_2$.

**Lemma 3.5.** *For any non-zero vectors $\theta_1, \theta_2 \in \mathbb{R}^d$,*

$$1 - \cos(\theta_1, \theta_2) \leq \frac{1}{2} \frac{\|\theta_1 - \theta_2\|_2^2}{\|\theta_1\|_2 \|\theta_2\|_2}. \tag{5}$$

With this result, we see that it suffices to upper bound $\cos(\theta_1, \theta_2)$ by $1 - \lambda_0 \langle w^{(1)} - w^{(2)}, \frac{\theta_1}{\|\theta_1\|_2} \rangle$. To develop this bound, let $\tilde{\theta}_i = \frac{\theta_i}{\|\theta_i\|_2}$ for $i = 1, 2$. The angle between $\theta_1$ and $\theta_2$ is the same as the angle between the normalized vectors $\tilde{\theta}_1$ and $\tilde{\theta}_2$. To calculate the cosine of the angle between $\tilde{\theta}_1$ and $\tilde{\theta}_2$, let $P$ be a plane spanned by $\tilde{\theta}_1$ and $w^{(1)} - w^{(2)}$ and passing through $w^{(1)}$ ($P$ is uniquely determined if $\tilde{\theta}_1$ is not parallel to $w^{(1)} - w^{(2)}$; if there are multiple planes, just pick any of them). Further, let $\hat{\theta}_2 \in \mathbb{S}^{d-1}$ be the unit vector along the projection of $\tilde{\theta}_2$ onto the plane $P$, as indicated in Fig. 1. Clearly, $\cos(\tilde{\theta}_1, \tilde{\theta}_2) \leq \cos(\tilde{\theta}_1, \hat{\theta}_2)$.

Consider a curve $\gamma(s)$ on $\mathrm{bd}(\mathcal{W})$ connecting $w^{(1)}$ and $w^{(2)}$ that is defined by the intersection of $\mathrm{bd}(\mathcal{W})$ and $P$ and is parametrized by its curve length $s$ so that $\gamma(0) = w^{(1)}$ and $\gamma(l) = w^{(2)}$, where $l$ is the length of the curve $\gamma$ between $w^{(1)}$ and $w^{(2)}$. Let $u_{\mathcal{W}}(w)$ denote the outer normal vector to $\mathcal{W}$ at $w$ as before, and let $u_\gamma : [0, l] \to \mathbb{S}^{d-1}$ be such that $u_\gamma(s) = \hat{\theta}$ where $\hat{\theta}$ is the unit vector parallel to the projection of $u_{\mathcal{W}}(\gamma(s))$ on the plane $P$. By definition, $u_\gamma(0) = \tilde{\theta}_1$ and $u_\gamma(l) = \hat{\theta}_2$. Note that in fact $\gamma$ exists in two versions since $\mathcal{W}$ is a compact convex body, hence the intersection of $P$ and $\mathrm{bd}(\mathcal{W})$ is a closed curve. Of these two versions we choose the one that satisfies that $\langle \gamma'(s), \tilde{\theta}_1 \rangle \leq 0$ for $s \in [0, l]$.[5] Given the above, we have

$$\cos(\tilde{\theta}_1, \hat{\theta}_2) = \langle \hat{\theta}_2, \tilde{\theta}_1 \rangle = 1 + \langle \hat{\theta}_2 - \tilde{\theta}_1, \tilde{\theta}_1 \rangle = 1 + \Big\langle \int_0^l u'_\gamma(s)\, \mathrm{d}s, \tilde{\theta}_1 \Big\rangle = 1 + \int_0^l \langle u'_\gamma(s), \tilde{\theta}_1 \rangle\, \mathrm{d}s. \tag{6}$$

Note that $\gamma$ is a planar curve on $\mathrm{bd}(\mathcal{W})$, thus its curvature $\lambda(s)$ satisfies $\lambda(s) \geq \lambda_0$ for $s \in [0, l]$. Also, for any $w$ on the curve $\gamma$, $\gamma'(s)$ is a unit vector parallel to $P$. Moreover, $u'_\gamma(s)$ is parallel to $\gamma'(s)$ and $\lambda(s) = \|u'_\gamma(s)\|_2$. Therefore,

$$\langle u'_\gamma(s), \tilde{\theta}_1 \rangle = \|u'_\gamma(s)\|_2 \langle \gamma'(s), \tilde{\theta}_1 \rangle \leq \lambda_0 \langle \gamma'(s), \tilde{\theta}_1 \rangle,$$

where the last inequality holds because $\langle \gamma'(s), \tilde{\theta}_1 \rangle \leq 0$. Plugging this into (6), we get the desired

$$\cos(\tilde{\theta}_1, \hat{\theta}_2) \leq 1 + \lambda_0 \int_0^l \langle \gamma'(s), \tilde{\theta}_1 \rangle\, \mathrm{d}s = 1 + \lambda_0 \Big\langle \int_0^l \gamma'(s)\, \mathrm{d}s, \tilde{\theta}_1 \Big\rangle = 1 - \lambda_0 \langle w^{(1)} - w^{(2)}, \tilde{\theta}_1 \rangle.$$

Reordering and combining with (5) we obtain

$$\langle w^{(1)} - w^{(2)}, \tilde{\theta}_1 \rangle \leq \frac{1}{\lambda_0} \Big(1 - \cos(\tilde{\theta}_1, \hat{\theta}_2)\Big) \leq \frac{1}{\lambda_0} \big(1 - \cos(\theta_1, \theta_2)\big) \leq \frac{1}{2\lambda_0} \frac{\|\theta_1 - \theta_2\|_2^2}{\|\theta_1\|_2 \|\theta_2\|_2}.$$

Multiplying both sides by $\|\theta_1\|_2$ gives (4), thus, finishing the proof. $\qquad \square$

**Example 3.6.** *The smallest principal curvature of some common convex bodies are as follows:*

- *The smallest principal curvature $\lambda_0$ of the Euclidean ball $\mathcal{W} = \{w \mid \|w\|_2 \leq r\}$ of radius $r$ satisfies $\lambda_0 = \frac{1}{r}$.*

- *Let $Q$ be a positive definite matrix. If $\mathcal{W} = \{w \mid w^\top Q w \leq 1\}$ then $\lambda_0 = \lambda_{\min} / \sqrt{\lambda_{\max}}$, where $\lambda_{\min}$ and $\lambda_{\max}$ are the minimal, respectively, maximal eigenvalues of $Q$.*

- *In general, let $\phi : \mathbb{R}^d \to \mathbb{R}$ be a $C^2$ convex function. Then, for $\mathcal{W} = \{w \mid \phi(w) \leq 1\}$, $\lambda_0 = \min_{w \in \mathrm{bd}(\mathcal{W})} \min_{v : \|v\|_2 = 1, v \perp \phi'(w)} \frac{v^\top \nabla^2 \phi(w) v}{\|\phi'(w)\|_2}$.*

In the stochastic i.i.d. case, when $\mathbb{E}[\Theta_t] = -\mu$, we have $\|\Theta_t + \mu\|_2 = O(1/\sqrt{t})$ with high probability. Thus say, for $\mathcal{W}$ being the unit ball of $\mathbb{R}^d$, one has $w_t = \Theta_t / \|\Theta_t\|_2$; therefore, a crude bound suggests that $\|w_t - w^*\|_2 = O(1/\sqrt{t})$, overall predicting that $\mathbb{E}[R_n] = O(\sqrt{n})$, while the previous result predicts that $R_n$ is much smaller. In the next example we look at the unit ball, to explain geometrically, what "causes" the smaller regret.

**Example 3.7.** *Let $\mathcal{W} = \{w \mid \|w\|_2 \leq 1\}$ and consider a stochastic setting where the $f_i$ are i.i.d. samples from some underlying distribution with expectation $\mathbb{E}[f_i] = \mu = (-1, 0, \ldots, 0)$ and $\|f_i\|_\infty \leq M$. It is straightforward to see that $w^* = (1, 0, \ldots, 0)$, and thus $\langle w^*, \mu \rangle = -1$. Let $E = \{-\theta \mid \|\theta - \mu\|_2 \leq \epsilon\}$. As suggested beforehand, we expect $-\mu_t \in E$ with high probability. As shown in Fig. 2, the excess loss of an estimate $\overrightarrow{OA}$ is $\langle \overrightarrow{OA}, \overrightarrow{OD} \rangle - 1 = |\tilde{B}D|$. Similarly, the excess loss of an estimate $\overrightarrow{OA'}$ in the figure is $|CD|$. Therefore, for an estimate $-\mu_t \in E$, the point $A$ is where the largest excess loss is incurred. The triangle $OAD$ is similar to the triangle $ADB$. Thus $\frac{|BD|}{|AD|} = \frac{|AD|}{|OD|}$. Therefore, $|BD| = \epsilon^2$ and since $|\tilde{B}D| \leq |BD|$, if $\|\mu_t - \mu\|_2 \leq \epsilon$, the excess error is at most $\epsilon^2 = O(1/t)$, making the regret $R_n = O(\log n)$.*

Our last result in this section is an asymptotic lower bound for the linear game, showing that FTL achieves the optimal rate under the condition that $\min_t \|\Theta_t\|_2 \geq L > 0$.

**Theorem 3.8.** *Let $h, L \in (0, 1)$. Assume that $\{(1, -L), (-1, -L)\} \subset \mathcal{F}$ and let $\mathcal{W} = \{(x, y) : x^2 + y^2/h^2 \leq 1\}$ be an ellipsoid with principal curvature $h$. Then, for any learning strategy, there exists a sequence of losses in $\mathcal{F}$ such that $R_n = \Omega\left(\log(n)/(Lh)\right)$ and $\|\Theta_t\|_2 \geq L$ for all $t$.*

### 3.2   Other regularities

So far we have looked at the case when FTL achieves a low regret due to the curvature of $\mathrm{bd}(\mathcal{W})$. The next result characterizes the regret of FTL when $\mathcal{W}$ is a polyhedron, which has a flat, non-smooth boundary and thus Theorem 3.3 is not applicable. For this statement recall that given some norm $\|\cdot\|$, its dual norm is defined by $\|w\|_* = \sup_{\|v\| \leq 1} \langle v, w \rangle$.

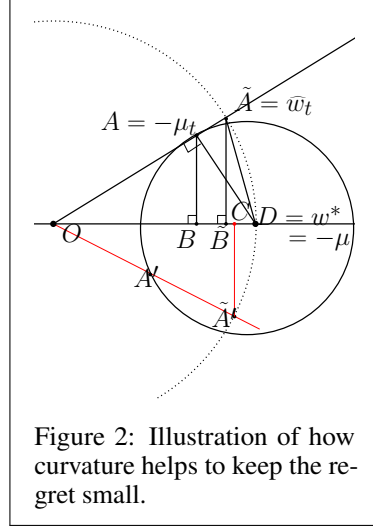

Figure 2: Illustration of how curvature helps to keep the regret small.

**Theorem 3.9.** *Assume that $\mathcal{W}$ is a polyhedron and that $\Phi$ is differentiable at $\Theta_i$, $i = 1, \ldots, n$. Let $w_t = \mathrm{argmax}_{w \in \mathcal{W}} \langle w, \Theta_{t-1} \rangle$, $W = \sup_{w_1, w_2 \in \mathcal{W}} \|w_1 - w_2\|_*$ and $F = \sup_{f_1, f_2 \in \mathcal{F}} \|f_1 - f_2\|$. Then the regret of FTL is*

$$R_n \leq W \sum_{t=1}^{n} t \, \mathbb{I}(w_{t+1} \neq w_t) \|\Theta_t - \Theta_{t-1}\| \leq FW \sum_{t=1}^{n} \mathbb{I}(w_{t+1} \neq w_t).$$

Note that when $\mathcal{W}$ is a polyhedron, $w_t$ is expected to "snap" to some vertex of $\mathcal{W}$. Hence, we expect the regret bound to be non-vacuous, if, e.g., $\Theta_t$ "stabilizes" around some value. Some examples after the proof will illustrate this.

*Proof.* Let $v = \mathrm{argmax}_{w \in \mathcal{W}} \langle w, \theta \rangle$, $v' = \mathrm{argmax}_{w \in \mathcal{W}} \langle w, \theta' \rangle$. Similarly to the proof of Theorem 3.3,

$$\langle v' - v, \theta' \rangle = \langle v', \theta' \rangle - \langle v', \theta \rangle + \langle v', \theta \rangle - \langle v, \theta \rangle + \langle v, \theta \rangle - \langle v, \theta' \rangle$$
$$\leq \langle v', \theta' \rangle - \langle v', \theta \rangle + \langle v, \theta \rangle - \langle v, \theta' \rangle = \langle v' - v, \theta' - \theta \rangle \leq W \, \mathbb{I}(v' \neq v) \|\theta' - \theta\|,$$

where the first inequality holds because $\langle v', \theta \rangle \leq \langle v, \theta \rangle$. Therefore, by Lemma 3.4,

$$R_n = \sum_{t=1}^{n} t \, \langle w_{t+1} - w_t, \Theta_t \rangle \leq W \sum_{t=1}^{n} t \, \mathbb{I}(w_{t+1} \neq w_t) \|\Theta_t - \Theta_{t-1}\| \leq FW \sum_{t=1}^{n} \mathbb{I}(w_{t+1} \neq w_t).$$

$\square$

As noted before, since $\mathcal{W}$ is a polyhedron, $w_t$ is (generally) attained at the vertices. In this case, the epigraph of $\Phi$ is a polyhedral cone. Then, the event when $w_{t+1} \neq w_t$, i.e., when the "leader" switches corresponds to when $\Theta_t$ and $\Theta_{t-1}$ belong to different linear regions corresponding to different linear pieces of the graph of $\Phi$.

We now spell out a corollary for the stochastic setting. In particular, in this case FTL will often enjoy a constant regret:

**Corollary 3.10** (Stochastic setting). *Assume that $(f_t)_{1 \le t \le n}$ is an i.i.d. sequence of random variables such that $\mathbb{E}[f_i] = \mu$ and $\|f_i\|_\infty \le M$. Let $W = \sup_{w_1, w_2 \in \mathcal{W}} \|w_1 - w_2\|_1$. Further assume that there exists a constant $r > 0$ such that $\Phi$ is differentiable for any $\nu$ such that $\|\nu - \mu\|_\infty \le r$. Then,*

$$\mathbb{E}[R_n] \le 2MW \left(1 + 4dM^2/r^2\right).$$

*Proof.* Let $V = \{\nu \mid \|\nu - \mu\|_\infty \le r\}$. Note that the epigraph of the function $\Phi$ is a polyhedral cone. Since $\Phi$ is differentiable in $V$, $\{(\theta, \Phi(\theta)) \mid \theta \in V\}$ is a subset of a linear subspace. Therefore, for $-\Theta_t, -\Theta_{t-1} \in V$, $w_{t+1} = w_t$. Hence, by Theorem 3.9,

$$\mathbb{E}[R_n] \le 2MW \sum_{t=1}^n \Pr(-\Theta_t, -\Theta_{t-1} \notin V) \le 4MW \left(1 + \sum_{t=1}^n \Pr(-\Theta_t \notin V)\right).$$

On the other hand, note that $\|f_i\|_\infty \le M$. Then

$$\Pr(-\Theta_t \notin V) = \Pr\left(\left\|\frac{1}{t} \sum_{i=1}^t f_i - \mu\right\|_\infty \ge r\right) \le \sum_{j=1}^d \Pr\left(\left|\frac{1}{t} \sum_{i=1}^t f_{i,j} - \mu_j\right| \ge r\right) \le 2de^{-\frac{tr^2}{2M^2}},$$

where the last inequality is due to Hoeffding's inequality. Now, using that for $\alpha > 0$, $\sum_{t=1}^n \exp(-\alpha t) \le \int_0^n \exp(-\alpha t)dt \le \frac{1}{\alpha}$, we get $\mathbb{E}[R_n] \le 2MW\left(1 + 4dM^2/r^2\right)$. $\square$

The condition that $\Phi$ is differentiable for any $\nu$ such that $\|\nu - \mu\|_\infty \le r$ is equivalent to that $\Phi$ is differentiable at $\mu$. By Proposition 2.1, this condition requires that at $\mu$, $\max_{w \in \mathcal{W}} \langle w, \theta \rangle$ has a unique optimizer. Note that the volume of the set of vectors $\theta$ with multiple optimizers is zero.

## 4 An adaptive algorithm for the linear game

While as shown in Theorem 3.3, FTL can exploit the curvature of the surface of the constraint set to achieve $O(\log n)$ regret, it requires the curvature condition and $\min_t \|\Theta_t\|_2 \ge L$ being bounded away from zero, or it may suffer even linear regret. On the other hand, many algorithms, such as the "Follow the regularized leader" (FTRL) algorithm, are known to achieve a regret guarantee of $O(\sqrt{n})$ even for the worst-case data in the linear setting. This raises the question whether one can have an algorithm that can achieve constant or $O(\log n)$ regret in the respective settings of Corollary 3.10 or Theorem 3.3, while it still maintains $O(\sqrt{n})$ regret for worst-case data. One way to design an adaptive algorithm is to use the $(\mathcal{A}, \mathcal{B})$-prod algorithm of Sani et al. [2014], leading to the following result:

**Proposition 4.1.** *Consider $(\mathcal{A}, \mathcal{B})$-prod of Sani et al. [2014], where algorithm $\mathcal{A}$ is chosen to be FTRL with an appropriate regularization term, while $\mathcal{B}$ is chosen to be FTL. Then the regret of the resulting hybrid algorithm $\mathcal{H}$ enjoys the following guarantees:*

- *If FTL achieves constant regret as in the setting of Corollary 3.10, then the regret of $\mathcal{H}$ is also constant.*

- *If FTL achieves a regret of $O(\log n)$ as in the setting of Theorem 3.3, then the regret of $\mathcal{H}$ is also $O(\log n)$.*

- *Otherwise, the regret of $\mathcal{H}$ is at most $O(\sqrt{n \log n})$.*

## 5 Conclusion

FTL is a simple method that is known to perform well in many settings, while existing worst-case results fail to explain its good performance. While taking a thorough look at why and when FTL can be expected to achieve small regret, we discovered that the curvature of the boundary of the constraint and having average loss vectors bounded away from zero help keep the regret of FTL small. These conditions are significantly different from previous conditions on the curvature of the loss functions which have been considered extensively in the literature. It would be interesting to further investigate this phenomenon for other algorithms or in other learning settings.

## Acknowledgements

This work was supported in part by the Alberta Innovates Technology Futures through the Alberta Ingenuity Centre for Machine Learning and by NSERC. During part of this work, T. Lattimore was with the Department of Computing Science, University of Alberta.

## Footnotes

[1] We let $\partial g(x)$ denote the subdifferential of a convex function $g : \mathrm{dom}(g) \to \mathbb{R}$ at $x$, i.e., $\partial g(x) = \left\{ \theta \in \mathbb{R}^d \,|\, g(x') \geq g(x) + \langle \theta, x' - x \rangle \; \forall x' \in \mathrm{dom}(g) \right\}$, where $\mathrm{dom}(g) \subset \mathbb{R}^d$ is the domain of $g$.

[2] The proofs not given in the main text can be found in the long version of the paper [Huang et al., 2016].

[3] $\mathbb{S}^{d-1} = \{ x \in \mathbb{R}^d \mid \|x\|_2 = 1 \}$ denotes the unit sphere in $d$-dimensions. All differential geometry concept and results that we need can be found in Section 2.5 of [Schneider, 2014].

[4]Following Schneider [2014], a convex body of $\mathbb{R}^d$ is any non-empty, compact, convex subset of $\mathbb{R}^d$.

[5]$\gamma'$ and $u'_\gamma$ denote the derivatives of $\gamma$ and $u$, respectively, which exist since $\mathcal{W}$ is $C^2$.

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
