[Reviews · NeurIPS 2016]

Reviewer 1

Summary

Follow the Leader (FTL) is a natural prediction algorithm that often works well in practice but typically has poor worst-case bounds in adversarial settings. This paper shows that one way of obtaining a low regret guarantee for FTL is to restrict the geometry of the class of predictors against which we compare. Preliminary simulations (in supplementary material) show that plain FTL indeed outperforms the regularised version (which has better worst-case guarantees) in some situations as predicted by the theory.

Qualitative Assessment

I really like the basic setting of this paper. It's interesting to ask what kind of realistic restrictions we can set to the adversary so that FTL would have a good regret bound. Looking at the geometry of the boundary of the constraint set is a quite novel approach into this. This is in contrast to more common approaches where low regret is obtained by assumptions about the loss function; the present paper considers the linear loss which is in some sense very difficult yet basic. The results are fairly promising and obtained using a variety of interesting techniques. The paper provides upper bounds for a couple of important scenarios, and a matching lower bound for one of them. I didn't read the proof of the lower bound (which constitutes the main part of the supplementary material), and I'm not familiar with all the mathematical tools used here, but with that reservation the proofs are clear and correct. There are also some preliminary simulation results in the supplementary material that support the theoretical predictions. The paper is very nicely written, with clear structure and motivation and an adequate exposition of some tools that might not be familiar to the average reader. (It would have been nice to have something about the experiments, and more discussion of some example scenarios, in the main paper instead the appendix, but also everything else in the paper is relevant and I can't really suggest anything that could be left out to accommodate the page limit, so I guess the author's made the right choice here.)

Confidence in this Review

2-Confident (read it all; understood it all reasonably well)


Reviewer 2

Summary

The paper studies the problem of regret minimization where the prediction in each time step is some point in convex region and the loss function at each time step is linear in this point vector. They show that the regret can be expressed in terms of a summmation involving the Bregmann divergence between the average value of the linear operator in the loss function over the previous time steps. From this they show that if the convex region has a high enough curvature then the regret is logarithmic. They show several examples where the curvature (max eigenvalue of the Hessian) is large.

Qualitative Assessment

Is it nice to see a strong characterization of the cases when the regret is logarithmic. However, I think the presentation can be improved. You should state early on the main result (perhaps Theorem 3.4) so that the main contribution is clear before reading through the proofs. Also the examples that you provide in are mostly specific shapes of convex bodies for W. But some motivating applications where such shapes arise would be better. Minor errors: Line 113: Fix "rather then bounded the regret"

Confidence in this Review

2-Confident (read it all; understood it all reasonably well)


Reviewer 3

Summary

This paper is about the performances of the Follow The Leader algorithm (FTL) in online linear optimization. While it is well known that is some settings FTL may incur linear regret, the authors show that if the optimization domain in nicely curved then FTL enjoys a logarithmic regret. A matching lower bound is provided. Some experiments are also carried out to support the theoretical findings. These results complement previous results about logarithmic regret when the regularity assumption was on the loss functions (e.g., strong convexity) instead of the optimization domain. They also mirror similar guarantees obtained by Levitin and Polyak (1966) for (non-sequential) convex optimization, as noted by the authors. The case when the optimization domain W is a polyhedron is also addressed: constant regret in shown in the i.i.d. setting under a restrcitive assumption on the support function of W.

Qualitative Assessment

First I would like to apologize for my late review. This paper is well written, easy to follow, and adds interesting new insights into the performances of the FTL algorithm. It should be of great interest to the NIPS community. Technical quality: - I must admit I am not an expert in differential geometry. From a high-level perspective the arguments seem correct but I cannot certify that Theorem 3.4 is valid under the exact assumptions of Theorem 3.4. Anyway the C^2 regularity assumption in the statement of Theoerm 3.4 is never explicitely used, which I think prevents the careful reader to check the proof precisely. Could the authors add rigorous justifications whenever conditions of this type are used within the proofs? - I do not agree with lines 49-52 in the introduction. Indeed if the loss vectors are i.i.d. [0,1]^d valued with E[\ell_{t,1}]=E[\ell_{t,2}]=1/4 and E[\ell_{t,j}]=3/4 for all j > 2, then any algorithm must suffer a regret on the simplex of \Omega(sqrt(n)). This is in the same spirit as any sqrt(T) lower bound in full information or bandit settings. (If you want to prove a sqrt(n) lower bound for the pseudo regret, you can take two arms with expected losses separated by 1/sqrt(n) and a random ordering for this two arms.) - If think there is a missing factor of 2 in the last bound of Theorem 3.11 (2*F insteaf of F) and in Corollary 3.13. Novelty: good. Significance: good. Clarity: the paper is globally well written. I however have (minor) comments: - I do not understand the role of Proposition 3.2: why isn't it used in the proof of Theorem 3.4? Cf. "it suffices to bound how much \Theta_t moves" in line 179. - Example 3.7 should be placed right after Theorem 3.4 to help illustrate it. - Could you add examples satisfying the strong assumption of Corollary 3.13, namely: \Phi differentiable on ball around \mu? Minor comments: - Since you do not assume W to be compact, the max defining the support function might be unattainable. So to be very rigorous it should be defined with a supremum instead. - Footenote 4: a convex body has a non-empty *interior*. - Proof of Theorem 3.4: I suggest to replace \theta_1, \theta_2, w_1, w_2 with \theta', \theta, v', v in order to better see the link with (3). - Corollary 3.13: please repeat here that W is a polyhedron.

Confidence in this Review

2-Confident (read it all; understood it all reasonably well)


Reviewer 4

Summary

After rebuttal ================ Reviewer 5 brings up a good point about the importance of the condition on the losses in Thm 3.4. It is important that this gets sufficiently emphasized in the final version of the paper, as the authors promise in their rebuttal, because the high-level message that fast rates *only* depend on the curvature of the constraint set is too simplistic. Original review ================ In online convex optimization, it is known that Follow-the-leader (FTL) achieves logarithmic regret for strongly convex losses, but FTL is generally considered unsuitable for linear losses, because it may then suffer linear regret. The present paper shows that, even for linear losses, FTL may be able to achieve logarithmic regret (for curved domains) or even finite regret (for polyhedral domains), provided that the average of the cost vectors stays away from 0. Technically, the results are based on Proposition 3.1, which rewrites the regret in terms of the differences between consecutive leaders, together with arguments showing that shape of the domain may cause the difference between leaders in consecutive rounds to be small.

Qualitative Assessment

The results of this paper are interesting for multiple reasons: 1) FTL is widely considered the most natural algorithm, it often works well in practice, and it is the standard algorithm in batch learning, so it seems strange that the online learning literature should be so dismissive of it. 2) As mentioned by the authors, it raises the question of whether the shape of the domain can also affect the regret rates for other methods. (Although regularized methods like gradient descent do not necessarily hit the boundary of the domain on every round like FTL, in high dimensions it does not seem unlikely that the domain constraints can be active quite frequently.) Comments: There is something that confuses me in the proof of Theorem 3.4: at the top of page 6 a plane P is constructed that is spanned by tilde{theta}_1 and w_2 - w_1, and passes through w_1. Under the assumptions of the theorem, I believe that w_i is simply a scaling of tilde{theta}_i. So then P is also spanned by tilde{theta}_1 and tilde{theta}_2 and passes through the origin. But then P also passes through tilde{theta}_2, so why is it necessary to consider the projection of tilde{theta}_2 onto P? I would suggest mentioning that the bound on the regret in terms of the number of leader changes in Theorem 3.11 is well-known, or at least follows easily from the be-the-leader lemma (for which you cite Lemma 2.1 of Shalev-Shwartz.) Minor comments: I wonder if Proposition 3.2 might extend to non-differentiable points Theta_t as well if the FTL choice w_t is suitably defined in case of ties. For instance, for the experts setting, I would guess that setting w_t to the uniform distribution on the set of leaders might do the trick. In general, the right definition might be arrived at by looking at Follow-the-Regularized leader and then letting the learning rate tend to infinity. Could you perhaps say something about why the lower bound on ||Theta_t|| is necessary for curved domains in Theorem 3.4, but not for polyhedral domains in Theorem 3.11? I guess this lower bound comes in in Theorem 3.4 in order to control the difference between w_{t+1} and w_t, right? So then in a sense the two results are analogous after all. Instead of requiring that Phi be differentiable in your results, I would suggest requiring that the FTL leader be unique, which seems easier to interpret. The fonts in Figure 2 are very small, making it hard to read.

Confidence in this Review

3-Expert (read the paper in detail, know the area, quite certain of my opinion)


Reviewer 5

Summary

The main question the authors try to answer is whether there are other types of regularities in Online Linear Optimization problem which allow the Follow-the-Leader (FTL) algorithm to achieve a small regret. They show that FTL can achieve an n-step regret of about (log n)/(L\lambda_0) if (1) the decision set has a smooth C^2-boundary with principal curvatures uniformly lower-bounded by some positive constant \lambda_0 and (2) the norms of time-averaged loss vectors are uniformly lower-bounded by some positive constant L. A lower bound of the same order is also shown for the case with the unit ball as the decision set. Moreover, the authors show that for polyhedral decision sets, which do not have condition (1), a constant expected regret can be achieved when loss vectors are stochastically generated. Finally, the authors mention that while FTL can have linear regret in general, one can use an existing approach to combine it with other algorithms, such as follow-the-regularized-leader, to guarantee a sublinear regret for worst-case scenarios.

Qualitative Assessment

The main point which the authors try to sell seem to be that the shape of the decision set plays a key role in the achievable regret bound, while previous works have only focused on the shapes of loss functions. However, we feel that the results in this paper may again say more about loss functions, instead of decision sets. Consider Theorem 3.4, which is the only result involving the shape of decision sets. It appears to us that the assumption \min_t \|\Theta_t\|_2 \ge L about the time-averaged loss vectors plays a bigger role, as it forces the loss vectors to point towards certain directions which makes possible for FTL to have small regret, while the curvature of decision sets seems to play a secondary role. I do not see how the curvature condition alone can ensure a small regret for FTL. Perhaps the authors can provide some intuition about this. Furthermore, the lower bound also demonstrates the effect of L, about loss functions, instead of the curvature of decision set, and so does the upper bound for polyhedral decision sets, as the small regret is guaranteed by the distribution of loss functions. Therefore, I feel that the results of this paper are still about loss functions as in previous works. If the authors want to emphasize the role of the decision sets, perhaps they could prove a regret lower bound in terms of their curvature. As for the originality of this paper we appreciate the authors’ findings of new types of regularities, both for decision set boundary and for loss vectors. However, it is not clear how broadly applicable such conditions are, as the authors did not provide any such discussion. In particular, it is not clear if decision sets typically have such curvature lower bound. The paper looks technically sound. Some proofs seem highly non-trivial, but at the same time are very hard to follow. It would be nice if the authors can provide more intuition and explanation before going into details. For example, the lower bound proof looks like a long stream of tedious calculation without any interesting high-level idea, and I wonder if there is a much simpler proof.

Confidence in this Review

2-Confident (read it all; understood it all reasonably well)


Reviewer 6

Summary

The paper basically investigatives regularities under which Follow The Leader (FTL) has good performance in online learning problems. It first mentions that even though FTL has linear regert in worst case, it exhibits small (logarithmic) regret in "curved" losses (e.g. exp-concave loss). However, instead of exploring loss functions, the paper mainly focuses on regularities in the curvature of the boundary of the domain (i.e. the non-empty, closed convex subset from which the "experts" are drawn) in which FTL has small regret. Using techniques in differential geometry, the paper proves small logarithmic regret (Theorem 3.4) for FTL in terms of parameters depending on curvature of the boundary of the domain, upper bounds on loss vectors, and lower bounds on support functions. Futhermore, the paper also seems to show finite regerts for FTL in case of stochastic or IID data and polyhedral domain. In addition, in the end, using (FTL, FTRL)-Prod [Sani et. al. 2014], the paper claims a meta-algorithm which has comparable regrets both in worst case and nice data scenario.

Qualitative Assessment

The flow of the paper seems unsatisfactory in the eyes of the reviewer. There are several results and discussions going on in the paper; however, they are not well separated and organized. For instance, the section 3 perhaps needs to be broken into several sub-sections exploring different parts of the results. Concretely, it looks a change of topic has happened at lines 222, 236 and 268. The paper looks to have acceptable analysis and technical quality when it establishes small regret results in the case of curvature regularities. This paper seems to be the first work exploring the regularities such as curvature of the boundary of the domain in online learning -- which is fairly new. In addition, this can also establish an impact of putting more emphasis on FTL algorithms again -- at least in cases where the adversary is not strong.

Confidence in this Review

1-Less confident (might not have understood significant parts)